# Study of Changes in the Ulan Buh Desert under the Dual Impacts of Desert Farmland Development and Climate Change

**DOI:** 10.3390/plants12193510

**Published:** 2023-10-09

**Authors:** Yujie Yan, Junyu Zhou, Wei Feng, Xinle Li, Zhiming Xin, Jin Xie, Jiaju Xi, Yiben Cheng

**Affiliations:** 1School of Soil and Water Conservation, Beijing Forestry University, Beijing 100083, China; yanyujie323@163.com (Y.Y.); zhoujunyu115@163.com (J.Z.); xzmlkn@163.com (Z.X.); 2Department of Grass and Livestock, Xilingol Vocational College, Xilinhot 026000, China; fw350@163.com; 3The Sand Forestry Experimental Center, Chinese Academy of Forestry, Dengkou, Bayannur 015200, China; nxylxl@126.com; 4National Meteorological Centre, China Meteorological Administration, Beijing 100081, China; xiej@cma.gov.cn; 5Department of Remote Sensing and Mapping, Space Star Technology Co., Ltd., Beijing 100086, China; xijj_cast503@sina.com

**Keywords:** Ulan Buh desert, desertification, vegetation coverage change, climate change, trend analysis

## Abstract

Desert farmland provides food for desert areas, but water is the main limiting factor of this region, thus desert farmland has an extremely fragile ecological environment. This study investigated the temporal and spatial variations of vegetation NDVI (Normalized Difference Vegetation Index) in the Ulan Buh Desert, China, from 1990 to 2022, using long-term Landsat satellite data obtained from the Google Earth Engine platform and local statistical data. The results showed that from 1990 to 2022, the NDVI exhibited relatively small fluctuations and a steady increase. Furthermore, the study analyzed the impact of climate factors, namely precipitation and temperature, on NDVI, and collected the groundwater lever changes under irrigation and farmland development. The results demonstrated a positive correlation between NDVI and both precipitation and temperature from 1990 to 2006. The study area experienced an overall trend of increasing humidity. Specifically, from 1990 to 2006, significant positive correlations with precipitation and temperature were observed in 4.4% and 5.5% of the region, respectively. From 2007 to 2022, significant positive correlations were observed in 5.4% and 72.8% of the region for precipitation and temperature, respectively. These findings suggest that temperature has become increasingly influential on vegetation NDVI, while the impact of precipitation remains relatively stable. Moreover, the study assessed the impact of human activities on vegetation NDVI. The results revealed that from 1990 to 2006, human activities contributed to 43.1% of the promotion of local vegetation NDVI, which increased to 90.9% from 2007 to 2022. This study provides valuable insights into the dynamics of vegetation in the Ulan Buh Desert and its response to climatic changes and human activities. The findings highlight the significance of climate conditions and human interventions in shaping the vegetation dynamics in the region, offering essential information for ecological restoration and conservation efforts.

## 1. Introduction

Vegetation is a sensitive indicator of climate change and the impacts of human activities, playing a crucial role in influencing ecosystem services and providing vital support for both natural ecosystems and human livelihoods [1,2,3,4]. In arid regions, vegetation serves as a valuable ecosystem service by effectively inhibiting desertification processes [5]. The Vegetation Index is an important indicator for assessing ecosystem vulnerability and a significant phenological indicator. In water-limited regions such as arid and semi-arid areas, vegetation growth exhibits substantial seasonal and interannual variability [6]. NDVI, as a necessary factor for vegetation growth and development, is greatly influenced by climate [7]. It shows a stronger ability to estimate the coverage of sparse vegetation and is particularly suitable for the arid Ulan Buh Desert regions [8]. Compared to the Enhanced Vegetation Index (EVI), NDVI effectively mitigates issues arising from the band ratio format and reduces the impacts of cloud shadows [9], topography, and solar angle variations [10,11,12]. Monitoring vegetation can effectively assess vegetation cover dynamics in arid and semi-arid regions and study the mechanisms of its changes. This is of crucial significance for evaluating the quality of ecosystem environments and maintaining optimal ecosystem functionality [13,14]. The vulnerability of the ecological environment will also impact regional economic development [15].

As one of the significant deserts in Northwestern China, the Ulan Buh Deserts have experienced notable changes in their ecological environment due to the combined impacts of human activities and climate change in recent years [16]. In previous studies, scholars have delved into the influence of human activities on the region, particularly focusing on agricultural development and irrigation, and their effects on the Normalized Difference Vegetation Index (NDVI) in the area. A series of research findings indicate that human activities have positively promoted the NDVI of the Ulan Buh Deserts [17]. With the implementation of ecological restoration projects such as wind–sand source control, cropland conversion to forests, grazing land conversion to grasslands, and ecological enclosures in Inner Mongolia since 1998, the NDVI of the deserts has demonstrated a significant upward trend. Notably, since 2007, the NDVI has increased at an annual rate of 0.0015, demonstrating the effectiveness of conservation measures. Furthermore, the impact of human activities on the NDVI has been continually increasing, accounting for 90.9% of the total area in recent years. This indicates that the positive effects of human activities in the region have been strengthened, providing crucial support for the vegetation recovery in the Ulan Buh [18,19,20]. On the other hand, climate change also serves as a crucial driving factor for the fluctuations in NDVI in the region. Precipitation and temperature are the primary climate factors influencing vegetation growth. Research has revealed that both precipitation and NDVI exhibit a positive correlation, while temperature shows a more pronounced impact on NDVI. Over the past few decades, the climate conditions in the region have demonstrated a trend of increasing humidity, with a gradual rise in precipitation [21,22]. These climate changes have had a positive effect on vegetation growth and recovery, making them essential contributors to the rising NDVI [23].

In conclusion, the Ulan Buh Deserts have shown a clear upward trend in NDVI under the combined influence of human activities and climate change. The proactive promotion by human activities and the improvement in climate conditions have provided significant impetus and support for the ecological recovery and vegetation growth in the region. However, with the continuous global climate change and ongoing human activities, the ecological environment of the Ulan Buh still faces challenges. Therefore, future research should explore more comprehensive and in-depth mechanisms of impact to formulate more scientific strategies for ecological conservation and management. In this study, we use NDVI data obtained from Google Earth Engine, and employ Sen’s slope analysis, correlation analysis, Hurst index analysis, and residual analysis to investigate the response of the Ulan Buh Desert to climate factors, future trends, and the influence of human activities. This research provides scientific evidence for long-term ecological conservation. Effective monitoring of vegetation dynamics is crucial for strengthening warning systems and assessing risks related to drought, desertification, and other natural disasters. This study employs Sen–MK trend analysis on NDVI, precipitation, and temperature data to explore temporal trends and their significance from 1990 to 2022. Additionally, it scrutinizes the correlations and significance between NDVI and precipitation as well as temperature. To predict forthcoming trends in vegetation cover, a Hurst analysis is conducted. Lastly, residual analysis is executed to assess the positive and negative influences of human activities on vegetation cover. The primary goal of this study is to offer valuable insights for the future sustainable development of agricultural land, the implementation of measures to mitigate sandstorms, and the enhancement of policies aimed at safeguarding and restoring the natural ecological environment. Furthermore, it aims to bolster warning systems and facilitate the assessment of drought and sandstorm-related risks through the efficient monitoring of precipitation stress and associated vegetation dynamics.

## 2. Results

### 2.1. Temporal and Spatial Variations in NDVI

#### 2.1.1. Interannual Variation Characteristics

As shown in Figure 1, the slope of NDVI in the Ulan Buh Desert remained relatively constant during 1990–2006. From the changes in NDVI values, it can be observed that NDVI showed a stable and slow upward trend during 1990–2006, with an R^2^ value of 0.176. In contrast, during 2007–2022, the vegetation NDVI exhibited an increasing trend with a slope of 0.0015 per year (a^−1^), and the R^2^ value was 0.6172, indicating a more pronounced upward trend compared to the period from 1990 to 2006. Overall, the NDVI of the Ulan Buh Desert showed an increasing trend, indicating vegetation recovery. The study period spanned from 1990 to 2022, and the year 2007 was chosen as the midpoint to compare the first 17 years with the latter 16 years.

#### 2.1.2. Spatial Variation Characteristics

As shown in Figure 2, during 1990–2006, the increasing NDVI areas accounted for 34.96% of the total area, with significantly increasing areas accounting for 5.26%, mainly distributed along the Yellow River Basin. The area of the Ulan Buh Desert with decreasing NDVI accounted for 28.96% of the total area, among which the significantly decreasing areas accounted for 18.61% and were mainly distributed in the central part of the desert. During this period, NDVI changes were dominated by stability and increase, showing the spatial variation characteristics of ‘stability in the middle and external growth’.

During 2007–2022, the area of the Ulan Buh Desert with decreasing NDVI accounted for 14.05% of the total area, among which significantly decreasing areas accounted for 12.06%, mainly distributed in the southern part of the desert. The stable NDVI areas accounted for 70.81% of the total area, while the increasing NDVI areas accounted for 15.14%, with significantly increasing areas accounting for 12.56%, which was more evident in the northeastern remote areas of the Ulan Buh Desert. This indicated that the areas with significant NDVI decrease gradually reduced, while the areas with significant NDVI increase increased. The proportion of stable NDVI areas increased significantly, indicating an overall increase in NDVI stability. Figure 3b shows a significant increase in the extent of the stable area in the study area, indicating that local ecological conservation measures are gradually taking effect. This suggests that the vegetation in the area is relatively stable, experiencing fewer disturbances or changes, and benefits from suitable soil, climatic conditions, and enhanced plant adaptability.

### 2.2. Influence of Climate Factors on NDVI

#### 2.2.1. Interannual Variations in Climate Factors

As shown in Figure 3, the precipitation showed a fluctuating upward trend with a rate of change of 1.57 mm·a^−1^ during 1990–2022, while the temperature remained relatively stable with a rate of change of 0.022 °C·a^−1^. In the Ulan Buh Desert, where temperature fluctuations were small and precipitation increased, this promoted the increase in NDVI by providing favorable conditions for vegetation growth. Figure 3 employs a comprehensive approach to investigate the changes in NDVI in relation to the total precipitation and annual mean temperature from 1990 to 2022. This analysis is conducted from both a holistic and partial perspective. Overall, the NDVI variations from 1990 to 2022 exhibit similarities with the patterns of precipitation changes.

**Figure 3 plants-12-03510-f003:**
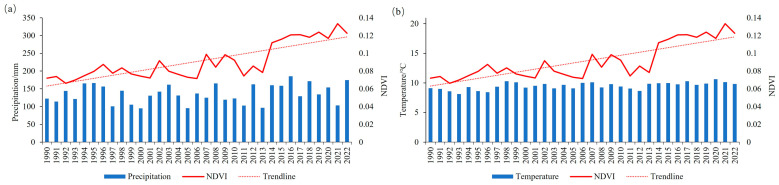
Plots of precipitation and temperature versus interannual variation in NDVI. Where (**a**) represents the plot of interannual variation of temperature versus NDVI; (**b**) represents the plot of temperature versus NDVI.

#### 2.2.2. Relationship between Climate Factors and NDVI

As shown in Figure 4 and Figure 5, the correlation between precipitation and NDVI was non-significantly positive during 1990–2006 and 2007–2022, with R values of 0.37 and 0.32, respectively. However, over the entire period from 1990 to 2022, the correlation became significantly positive with an R value of 0.38. During 1990–2006, the significantly positive correlation accounted for 4.4%, while the non-significantly positive correlation accounted for 42.90% (Table 1). In contrast, during 2007–2022, the significantly positive correlation accounted for 5.4%, while the non-significantly positive correlation accounted for 90.2%. The spatial distribution of the correlation between precipitation and NDVI remained relatively stable over the years and was mainly characterized by a non-significantly positive relationship.

The connection between temperature and NDVI was not significantly positive during 1990–2006 and 2007–2022, with R values of 0.29 and 0.57, respectively. However, over the entire period from 1990 to 2022, the correlation became significantly positive with an R value of 0.56. During 1990–2006, the significantly positive correlation accounted for 5.5%, while the non-significantly positive correlation accounted for 42.90%. In contrast, during 2007–2022, the significantly positive correlation accounted for 72.8%, while the non-significantly positive correlation accounted for 24.7%. This indicates that the relationship between temperature and NDVI gradually shifted from a non-significantly positive correlation to a significantly positive correlation, and the impact of temperature on NDVI gradually increased. The significantly positive correlation was sparsely distributed in the northern region, but later it increased significantly in the southern region. This suggests that, in addition to the increased precipitation, the stability of temperature also helps to maintain vegetation moisture and promote vegetation growth.

### 2.3. Future Trend of Vegetation

Hurst index analysis was conducted to predict the future trend of NDVI in the Ulan Buh Desert. The average *H* value for the entire region was 0.4819, with *H* < 0.5, indicating that the overall trend is expected to remain relatively stable in the future, showing a slight reverse trend compared to the present. Based on the NDVI trend grid data and *H* index grid data, the future trend of NDVI was divided into four categories: continuous improvement, continuous degradation, future improvement, and future degradation. Among them, the continuous improvement area accounted for 24.8%, the continuous degradation area accounted for 18.5%, the future improvement area accounted for 31.1%, and the future degradation area accounted for 23.5% (Table 2).

As shown in Figure 6, the main concentration of continuous improvement in the future NDVI trend was observed along the Yellow River Basin, indicating the positive influence of the Yellow River Basin on future vegetation improvement. The future improvement of NDVI is mainly concentrated in the east, the future degradation is mainly concentrated in the west, and there is a partial continuous degradation trend in the central and southeast.

### 2.4. Future Trend of Vegetation

As shown in Figure 7, during 1990–2006, human activities promoted NDVI mainly in the southern, eastern, and western margins of the desert, while human activities inhibited NDVI mainly in the central and northern parts. In contrast, during 2007–2022, human activities promoted NDVI in the entire study area, while human activities inhibited NDVI mainly along the Yellow River Basin. The spatial distribution of the impact of human activities on NDVI indicates an increasing promotion effect on vegetation growth and a wide range of impacts, covering almost the entire study area.

As shown in Table 3, during 1990–2006, human activities accounted for 43.1% of the promotion of NDVI, while during 2007–2022, this contribution increased to 90.9%. The proportion of promotion effect by human activities increased by 47.8%, covering an area of 5318.1 km^2^, while the inhibiting effect on NDVI decreased to 7.6% during 2007–2022, which was 48.2% less than during 1990–2006, with an area change of 5364.6 km^2^.

From 1990 to 2006, the influence of anthropogenic human activities and climate change on vegetation NDVI was 52.2% and 47.8%, respectively. From 2007 to 2022, the impact of human activities and climate change on vegetation NDVI was 90.4% and 9.6%, respectively. With the gradual increase in NDVI in the Ulan Buh Desert, human activities play a key role in protecting the local environment and promoting vegetation growth and restoration.

## 3. Materials and Methods

### 3.1. Study Area

The Ulan Buh Desert is one of the eight major deserts in China and one of the five major deserts in Inner Mongolia. It is located in the western part of Inner Mongolia, at the junction of North China and Northwest China, within parts of Alxa League, Bayannur City, and Wuhai City in the Inner Mongolia Autonomous Region. This region is ecologically fragile and belongs to the arid northwest region of China, situated at the forefront of the northwest desert and semi-desert [24,25]. Its geographical coordinates are in the range of 106°36′–107°07′ E and 39°17′–40°55′ N.

The Ulan Buh Desert lies in the Yellow River, and to the south is the Helan Mountain. The total area is approximately 10,000 square kilometers [24,26,27]. Previous studies have indicated that the Ulan Buh Desert and its neighboring regions contain various landscape types, such as deserts, Gobi areas, mountainous hills, saline–alkali lands, vegetation, water bodies, human settlements, and others [28]. The Ulan Buh Desert plays a vital role in maintaining the ecological security of the middle and lower reaches of the Yellow River Basin and has a significant impact on the sustainable development of the regional economy [29]. The boundary range of the study area is shown in Figure 8.

### 3.2. Datasets and Pre-Processing

The NDVI data used in this study were obtained from the cloud computing platform Google Earth Engine (GEE). The data type consists of a long-time series of Landsat 5/7/8 satellite remote sensing data with a spatial resolution of 30 m and a temporal resolution of 16 days. (Table 4) The NDVI has 416 data points. The primary algorithm utilized involves the use of Google Earth Engine (GEE) to select Landsat 5/7/8 images while effectively masking out cloud cover. Subsequent steps encompass the extraction of the red- and near-infrared bands from the selected images, the calculation of NDVI values, image processing, cropping to the specific study area, and the generation of visual representations of the NDVI data. The NDVI data for the growing season (April to October) from 1990 to 2022 were generated using the maximum synthesis method on a monthly scale. These data were further processed, including projection transformation, mosaicking, and cropping, to obtain the NDVI dataset for the Ulan Buh Desert and neighboring areas. Additionally, Landsat remote sensing images were utilized for cloud removal analysis. The meteorological data were sourced from the National Meteorological Science Data Center, accessible via http://data.cma.cn/ (accessed on 16 July 2023). This dataset includes climate factors such as atmospheric pressure, temperature, precipitation, air humidity, and wind speed. Daily precipitation and temperature data from meteorological stations in the Ulan Buh Desert area for the years from 1990 to 2022 were collected and processed into monthly scale data. Using ANUSPLIN software, average temperature and total annual precipitation for each year from 1990 to 2022 were interpolated. Both the average temperature and annual precipitation data have a spatial resolution of 1 km. The Digital Elevation Model (DEM) data used in this study were obtained from NASADEM_HGT, with a spatial resolution of 30 m. The time series for NDVI, precipitation, and temperature span from 1990 to 2022, covering 32 years of continuous data. The DEM data, on the other hand, represents only one year.

### 3.3. Methods

#### 3.3.1. Sen + Mann–Kendall Trend Analysis

The Sen + Mann–Kendall trend analysis is used for analyzing long-time series vegetation data and can be applied to analyze the multi-year variations of vegetation in the Ulan Buh Desert. The computation formulas are as follows:(1)$$\beta =median\left(\frac{x_{j}-x_{i}}{j-i}\right)\forall j>i$$
where the trend degree β is used to determine the rising or falling trend of a time series. When β is positive, NDVI shows an increasing trend over time, whereas when β is negative, the trend is the opposite.

Mann–Kendall trend analysis is a non-parametric statistical test method. For the time series $x_{i}$, where *i* = 1, 2, …, n, the standardized test statistic *Z* is defined as follows:(2)$$Z=\left\{\begin{array}{c} \frac{S-1}{\sqrt{Var\left(S\right)}}\left(S>0\right)\\ \;\;\;\;\;\;\;0\;\;\;\;\;\;\;\;\left(S=0\right)\\ \frac{S+1}{\sqrt{Var\left(S\right)}}\left(S<0\right) \end{array}\right.$$
(3)$$S=\sum_{i=1}^{n-1}\sum_{j=i+1}^{n}sgn(x_{j}-x_{i})$$
(4)$$sgn\left(\theta \right)=\left\{\begin{array}{c} 1,\;\theta >0\\ 0,\theta =0\\ -1,\theta <0 \end{array}\right.$$
(5)$$Var\left(S\right)=\frac{n(n-1)(2n+5)}{18}$$

Based on the standard, it was divided into 5 levels: significant increase (β > 0, *|Z|* ≥ 1.96), insignificant increase (β > 0, |*Z*| < 1.96), stable (β = 0, |*Z*| < 1.96), insignificant decrease (β < 0, |*Z|* < 1.96), and significant decrease (β < 0, *|Z|* ≥ 1.96).

The NDVI, precipitation, and temperature data for the periods 1990–2006 and 2007–2022 were inputted separately into the MATLAB Starter Application as raster datasets. The code was then executed to compute the corresponding trends and perform the Mann–Kendall test for each individual year. In the ArcGIS 10.8 software, masked extraction was performed using the obtained results. The trend maps were generated directly from the code’s output, while the significance results were categorized based on the trend maps. Eventually, the figures were produced. All the data underwent stationarity testing using the Augmented Dickey–Fuller (ADF) test within the SPSS software. The results of this testing confirm the stationarity of the data. To elaborate, after performing first-order differentiation on the NDVI data, its ADF statistic was −8.46366, with a *p*-value of 0.0078. This *p*-value is significantly smaller than the 5% significance level, providing robust evidence that the sequence is indeed stationary.

Regarding the ‘precipitation’ variable, its ADF statistic was −5.994203, with a *p*-value of 0.0000, strongly indicating stationarity since the *p*-value is very close to zero and significantly less than the 5% significance level. As for the ‘temperature’ variable, its ADF statistic was −3.218903, with a *p*-value of 0.0280. While the *p*-value is not extremely small, it is still less than the 5% significance level, suggesting that the sequence can be considered stationary. In summary, based on these results, it can be confidently asserted that the sequence exhibits strong stability after first-order differencing, establishing first-order integer stationarity.

#### 3.3.2. Correlation Analysis of NDVI with Various Factors

We assessed the connection between NDVI and various factors to determine the relationship between each variable factor and vegetation coverage (NDVI). The formula used for calculating the correlation is as follows:(6)$$R_{xy}=\frac{\sum_{i=1}^{n}\left[\left(x_{i}-\bar{x}\right)\left(y_{i}-\bar{y}\right)\right]}{\sqrt{\sum_{i=1}^{n}{\left(x_{i}-\bar{x}\right)}^{2}\sum_{i=1}^{n}{\left(y_{i}-\bar{y}\right)}^{2}}}$$
where $R_{xy}$ represents the correlation coefficient between different factors and NDVI. The $x_{i}$ and $I_{i}$ i represent the climate factor and NDVI values for the *i*th year, respectively. The $x_{i}$ and $y_{i}$ represent the average values of the climate factor and NDVI, respectively. Additionally, a *t*-test was employed to determine the significance of the correlation. In this study, the significance level for testing the correlation was set at 0.05. In the RStudio 4.4.2 software, the Excel-formatted data can be imported for NDVI, precipitation, and temperature for the periods 1990–2006 and 2007–2022. Prior to that, the necessary packages must be downloaded. By utilizing the correlation formula and specific code, a correlation scatter plot can be generated to visualize the relationships.

#### 3.3.3. Hurst Exponent Analysis

Hurst exponent analysis is effective in predicting the future trends of time series and finds wide application in fields such as ecology. The computation process follows.

For a time-series *NDVI(t)*, where *t* = 1, 2, 3, …, *n*, and any positive integer *p* ≥ 1, the mean sequence *NDVI(p)* is defined as follows:(7)$$NDVI\left(p\right)=\frac{1}{p}\sum_{t=1}^{p}NDVI\left(t\right)$$

Define the sequence of cumulative deviations *NDVI(t,p)* as:(8)$$NDVI\left(t,p\right)=\sum_{t=1}^{t}\left[NDVI\left(i\right)-NDVI\left(p\right)\right],1\leq t\leq p$$

Define the polar sequence *R(p)* to be:(9)$$R\left(p\right)=\underset{1\leq t\leq p}{\max}\left[NDVI\left(t,p\right)\right]-\underset{1\leq t\leq p}{\min}\left[NDVI\left(t,p\right)\right]$$

Define the standard deviation sequence *S(p)* to be:(10)$$S\left(p\right)=\sqrt{\frac{1}{p}\sum_{t=1}^{p}{\left[NDVI\left(t\right)-NDVI\left(p\right)\right]}^{2}}$$

Hurst summarizes the relationship through long practice as follows:(11)$$\frac{R\left(P\right)}{S\left(p\right)}={\left(\alpha p\right)}^{H}$$
where *H* represents the Hurst exponent. When *H* = 0.5, the NDVI time series is a random sequence. When 0.5 < *H* < 1, the NDVI time series exhibits positive persistence, indicating that the future NDVI trends will continue to be consistent with the past, and a larger *H* value indicates stronger persistence. Conversely, when 0 < *H* < 0.5, it indicates that the time series is anti-persistent, and a smaller *H* value indicates stronger anti-persistence. For the analysis of the Hurst index, this study analyzed 32 years of NDVI data, with 13 data points per year, for a total of 416 data points. In total, more than 256 data points were analyzed. The results are highly credible as the data points meet the quantitative requirements of the Crevecoeur et al. study [30]. To conduct the Hurst trend prediction, the MATLAB Starter Application required the loading of raster data for the years 1990–2022. Subsequently, the corresponding code was inputted to forecast future trends. These predicted trend images for the NDVI were then integrated, and classification and masked extraction were performed using ArcGIS 10.8 software. It combines the results of NDVI trend changes to categorize future trends into future prevention, future degradation, sustained improvement, and sustained degradation.

#### 3.3.4. Residual Analysis

This study assumes that NDVI is only influenced by human activities and climatic conditions and utilizes residual analysis to evaluate the impact of human activities on NDVI [31]. The calculation formula is shown below:(12)$$NDVI\left(pre\right)=a+bx_{1}+cx_{2}$$
(13)$$\sigma =NDVI\left(obs\right)-NDVI\left(pre\right)$$
where $NDVI\left(pre\right)$ represents the predicted NDVI value, *x*_1_ represents annual average temperature, *x*_2_ represents annual total precipitation, ‘*a*’ represents the constant term, ‘*b*’ and ‘*c*’ are the regression coefficients, ‘*σ*’ denotes the residual value, *NDVI(obs)* are the observed NDVI values. A positive residual trend indicates that human activities have a promoting effect on NDVI, while a negative trend indicates that human activities have an inhibiting effect on NDVI [32]. For residual analysis, the MATLAB Starter Application was utilized to load NDVI raster images corresponding to the periods 1990–2006 and 2007–2022. The code was executed to compute the predicted NDVI values, residuals, one-dimensional linear regression trends, and the impacts of climate change and human activities for each year. Ultimately, the processed raster images underwent masked extraction and classification using ArcGIS 10.8.

## 4. Discussion

The arid ecological vulnerability and ecosystem service value of the Ulan Buh Desert have shown a trend of deterioration followed by improvement since the 1990s, indicating an improvement in the current ecological conditions [33]. Since 1998, the Inner Mongolia region, to which the study area belongs, has undertaken ecological restoration initiatives, such as combating desertification in the Beijing–Tianjin sand source areas, converting farmland to forests, returning grazing land to grassland, conservation enclosure, and crop rotation and fallow [34]. These efforts have greatly improved the ecological environment. In the face of severe global climate change, the Ulan Buh Desert should continuously implement rational measures based on climatic conditions to reduce desertification [35,36].

In this study, the GEE platform was used to monitor the long-term dynamic changes in vegetation in the Ulan Buh Desert from 1990 to 2022 using Landsat remote sensing images. The findings indicated that the area with increased NDVI constituted 50.1% of the study area, suggesting an improvement in the vegetation and ecological environment. This finding is consistent with the results of previous studies conducted by Cheng Y., Yudong C., and others [37,38,39]. The research results also revealed that increased precipitation had a positive impact on local vegetation NDVI when the temperature remained stable, leading to a continuous increase in NDVI. This is in line with the findings of Xia P., Li Jun, and others [33,40]. Munyati C. et al. [41]. studied the relationship between vegetation cover and climate in Guizhou from 2001 to 2018, and the results indicated a correlation between vegetation cover and temperature and precipitation. The present study found that temperature had a greater impact on the increase in vegetation NDVI than precipitation. The study also predicted the future changes in NDVI, revealing a potential degradation trend overall, which may be owed to agricultural expansion. Human activities can bring about changes in vegetation [42,43,44]. The positive impact of human activities is continuously expanding, promoting local vegetation recovery and growth, which is consistent with the findings of Fengmin L. and others [45], and it is inseparable from proactive government policies [46,47,48]. We have compared the rate of NDVI increase between the Ulan Buh Desert and the Mongolian Plateau. The Mongolian Plateau witnessed substantial growth from 2010 to 2019, exhibiting a regional increase of 9.08% in comparison to the period from 2000 to 2009. Conversely, in the Ulan Buh Desert, a significant increase of 7.30% occurred between 2007 and 2022 when compared to the period from 1990 to 2006. Overall, the growth rate of the Ulan Buh Desert and the plateau is relatively gradual. The disparity between the NDVI trend of the Ulan Buh Desert and the predicted *H* value of the Mongolian Plateau is only 0.0119, suggesting a consistent future trend. The research area exhibits correlations with both precipitation and temperature, while the Mongolian Plateau is only correlated with precipitation.

This table presents soil moisture data at varying depths spanning from 1990 to 2022. The dataset facilitates an in-depth examination of the long-term fluctuations in soil moisture, particularly considering the impacts stemming from agricultural irrigation practices and alterations in vegetation cover. As shown in Table 5, the volumes of the first to fourth layers of groundwater in the desert regions have remained relatively stable with a slight decreasing trend from 1990 to 2022. This indicates the need for greater attention to local farmland management, and the adoption of more suitable drip irrigation methods to rationalize water resource allocation. Based on on-site experimental data observations, different types of areas exhibited varied annual fluctuations in groundwater levels from 2006 to 2016 (Figure 9). Specifically, the well-irrigated area showed a slight increase, while the yellow river-irrigated area and desert area experienced a slight decrease. The groundwater depth in the farmland irrigation area fluctuated but remained stable overall.

Expanding the Scope of Climate Factors: this study primarily concentrated on two climate factors, namely annual mean temperature and annual total precipitation, when assessing their influence on NDVI vegetation. Future research endeavors should encompass a broader range of meteorological factors. Additionally, it is essential to integrate land-use categories and examine NDVI variations under diverse land cover scenarios. Subsequent studies should also take into account local policies, harmoniously blending theory and practice, and provide a thorough exploration of how vegetation responds to both climate and human activities.

## 5. Conclusions

In 2023, the frequency of sandstorms in the Ulan Buh Deserts continues to increase. Our objective is to uncover the key environmental changes occurring in this region. This study provides an in-depth analysis of the changes in NDVI and its influencing factors, while also examining the future trends of NDVI and the impact of human activities.

The Ulan Buh Deserts are unique deserts with contrasting features. On one hand, they are characterized by extremely dry and arid meteorological conditions, leading to frequent sandstorms. On the other hand, their location along the Yellow River has allowed them to benefit from extensive irrigation through the Yellow River water projects. Consequently, large areas of farmland have been developed, making this region an extraordinary desert agricultural area. Under the dual influence of human activities and climate change, the Ulan Buh Deserts have exhibited notable transformations. The study results are as follows:From 1990 to 2006, the vegetation NDVI in the study area showed relatively small fluctuations and remained stable, with the increased area accounting for 34.96% of the total study area. In 2007 to 2022, the vegetation NDVI increased at a rate of 0.0015 per year, with the stable and increased areas accounting for 70.81% and 15.14% of the total area, respectively. This indicates that the changes in climatic conditions have promoted the growth and recovery of local vegetation.From 1990 to 2022, both precipitation and temperature had a positive correlation with NDVI, and the climate in the study area exhibited a trend of increasing humidity. From 1990 to 2006, the proportion of significant positive correlation between NDVI and precipitation was 4.4%, and the proportion between NDVI and temperature was 5.5%. From 2007 to 2022, the proportions of significant positive correlation between NDVI and precipitation and temperature were 5.4% and 72.8%, respectively. This indicates that the influence of temperature on vegetation NDVI increased during this time period, while the impact of precipitation remained relatively stable.From 1990 to 2022, future trend predictions for the study area indicate relative stability but with a downward trajectory. This suggests that humans should increase positive activities and efforts to protect vegetation, especially in the agricultural activities primarily focused on farmland. It is essential to formulate more scientifically based policies under the Yellow River water projects to increase the area for future improvement and sustainable enhancement.From 1990 to 2006, human activities contributed to 43.1% of the promotion of local vegetation NDVI, while from 2007 to 2022, human activities contributed to 90.9% of the promotion of local vegetation NDVI, with their contribution continuously increasing. This indicates that positive human activities have a strong promoting effect on vegetation growth and recovery.

## Figures and Tables

**Figure 1 plants-12-03510-f001:**
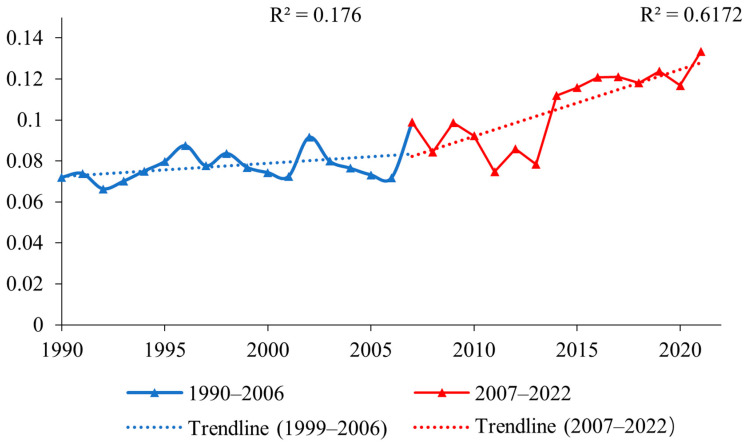
Changes in NDVI trends in the Ulan Buh Desert for 1990–2022. The blue line represents the changes in NDVI trends of 1990–2006, The blue line represents the changes in NDVI trends of 2007–2022.

**Figure 2 plants-12-03510-f002:**
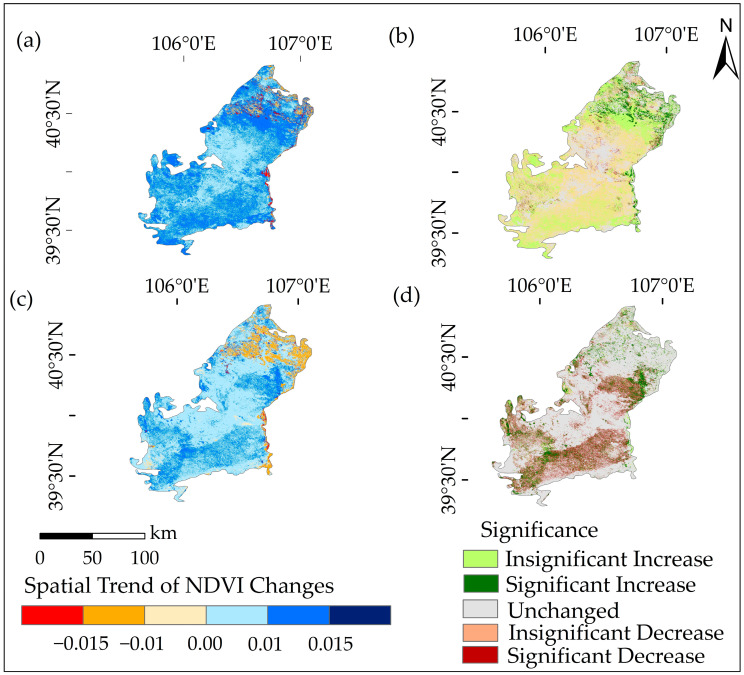
Trends and tests of vegetation growth in the Ulan Buh Desert. Where (**a**) represents the trend of NDVI in 1990–2006; (**b**) represents the test of NDVI in 1990–2006; (**c**) represents the trend of NDVI in 2007–2022; and (**d**) represents the test of NDVI in 2007–2022.

**Figure 4 plants-12-03510-f004:**
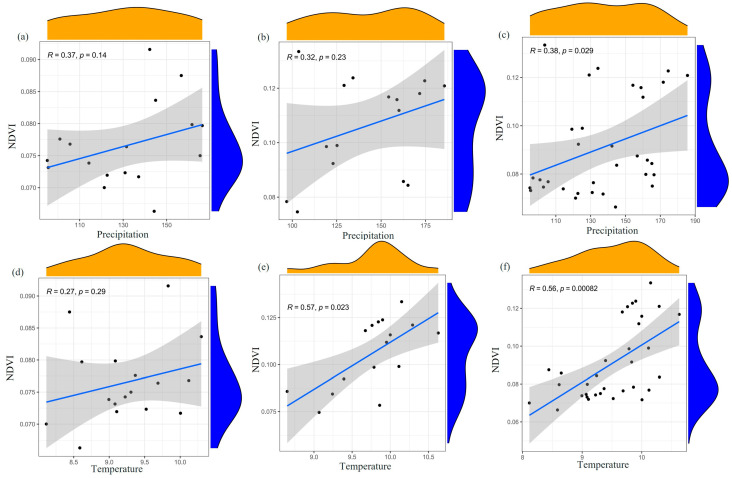
Scatterplot of the correlation between climate factors and NDVI, where (**a**) represents the correlation relationship between annual total precipitation and NDVI for 1990–2006; (**b**) represents the correlation relationship between annual total precipitation and NDVI for 2007–2022; (**c**) represents the relationship between annual total precipitation and NDVI for 1990–2022; (**d**) represents the correlation relationship between annual average temperature and NDVI for 1990–2006; (**e**) represents the correlation relationship between annual average temperature and NDVI for 2007–2022; and (**f**) represents the correlation between annual average temperature and NDVI for 1990–2022.

**Figure 5 plants-12-03510-f005:**
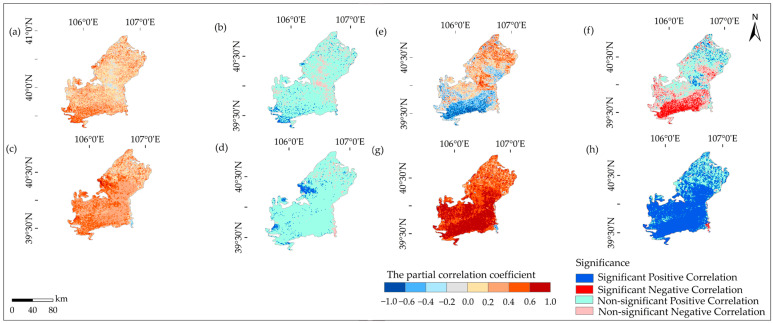
Coefficient of correlation and significant correlation plots of climate factors with NDVI; the left plot indicates the relationship between precipitation and NDVI, and the right plot indicates the relationship between temperature and NDVI. Where: (**a**,**e**) represent the correlation coefficient between climate factors and NDVI in 1990–2006, (**b**,**f**) represent the correlation coefficient between climate factors and NDVI in 1990–2006, (**c**,**g**) represent the correlation coefficient between climate factors and NDVI in 2007–2022, (**d**,**h**) represent the correlation coefficient between climate factors and NDVI in 1990–2006.

**Figure 6 plants-12-03510-f006:**
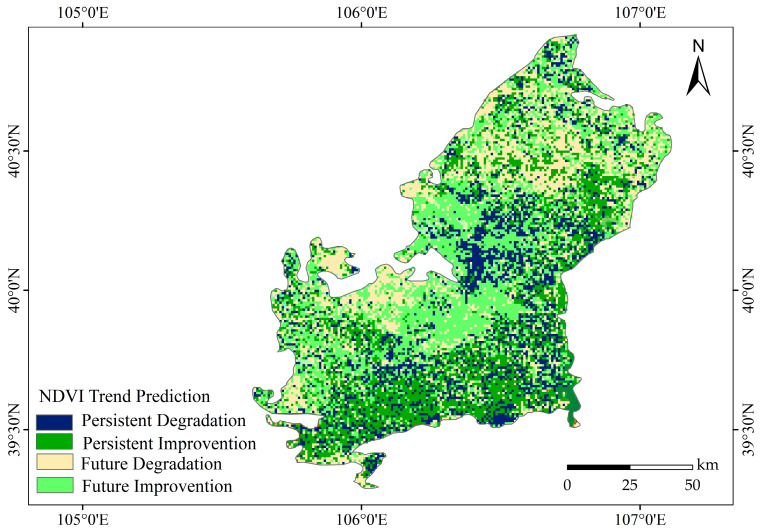
Spatial distribution of future trends of NDVI in the study area.

**Figure 7 plants-12-03510-f007:**
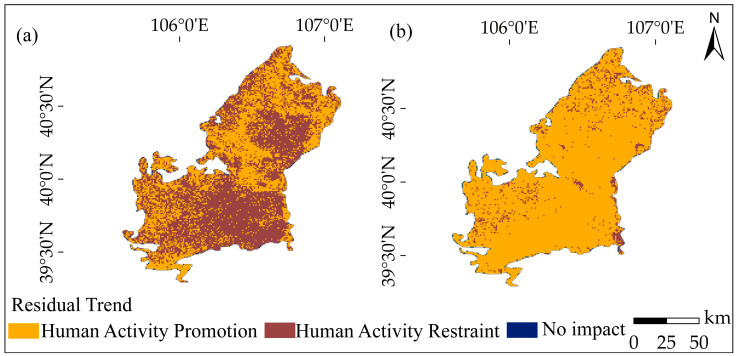
Spatial distribution of anthropogenic impacts on NDVI, where (**a**) represents the spatial distribution of anthropogenic impacts on NDVI from 1990–2006 and (**b**) represents the spatial distribution of anthropogenic impacts on NDVI from 2007–2022.

**Figure 8 plants-12-03510-f008:**
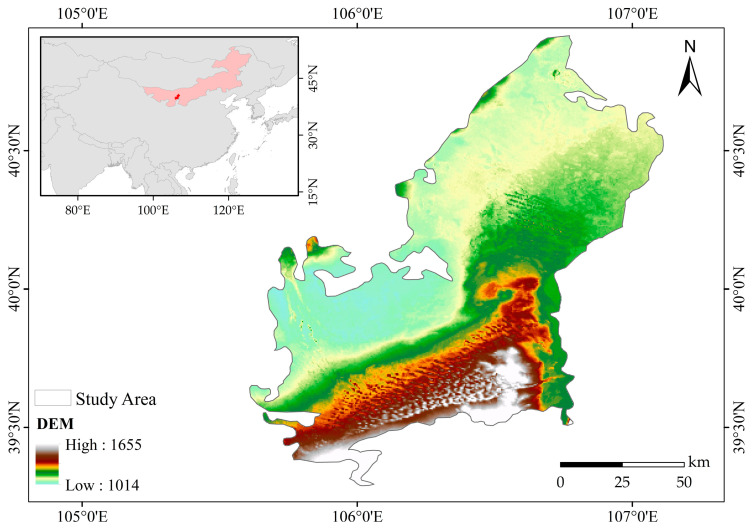
Study area overview.

**Figure 9 plants-12-03510-f009:**
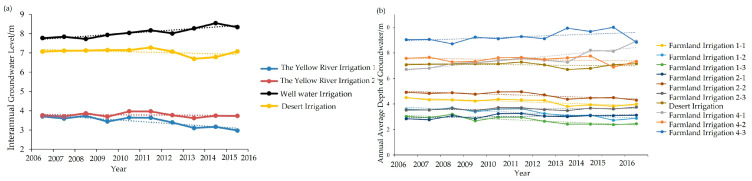
The distribution of groundwater in the study area from 2006 to 2016, where (**a**) represents the interannual groundwater level and (**b**) represents the annual average depth of groundwater.

**Table 1 plants-12-03510-t001:** Distribution of climate factor correlations.

Year	Climate Factors	Significant Positive Correlation		Non-Significant Positive Correlation		Significant Negative Correlation		Non-Significant Negative Correlation	
		Proportion/%	Area/km^2^	Proportion/%	Area/km^2^	Proportion/%	Area/km^2^	Proportion/%	Area/km^2^
1990–2006	Precipitation	4.4	490.1	82.8	9223.2	0.01	1.1	12.7	1414.7
2007–2022		5.4	601.5	90.2	10,047.5	001	1.1	4.3	479.0
1990–2006	Temperature	5.5	612.7	42.9	4778.7	15.6	1737.7	36.0	4010.1
2007–2022		72.8	8109.3	24.7	2751.4	0.05	5.7	2.0	222.8

**Table 2 plants-12-03510-t002:** Future trends in vegetation percentage and area.

Year	Persistent Prevention		Persistent Degradation		Future Prevention		Future Degradation	
	Proportion/%	Area/km^2^	Proportion/%	Area/km^2^	Proportion/%	Area/km^2^	Proportion/%	Area/km^2^
1990–2022	24.8	2762.5	18.5	2060.7	31.1	3464.3	23.5	2617.7

**Table 3 plants-12-03510-t003:** Role of anthropogenic activities by vegetation NDVI.

Year	Promotion		Restraint		No Impact	
	Proportion/%	Area/km^2^	Proportion/%	Area/km^2^	Proportion/%	Area/km^2^
1990–2006	43.1	4806.5	55.8	6215.7	1.1	117.0
2007–2022	90.9	10,124.6	7.6	851.1	1.5	163.5

**Table 4 plants-12-03510-t004:** Data sources and resolutions.

Data Type	Dataset	Spatial Resolution	Length of Time
NDVI (1990–2011)	LANDSAT/LT05/C02/T1_L2	30 m	32
NDVI (2012–2013)	LANDSAT/LE07/C02/T1_L2		
NDVI (2014–2022)	LANDSAT/LC08/C02/T1_L2		
Precipitation	http://data.cma.cn/ (accessed on 5 July 2023)	1 km	
Temperature	http://data.cma.cn/ (accessed on 5 July 2023)	1 km	
DEM	NASA/NASADEM_HGT/001	30 m	1

**Table 5 plants-12-03510-t005:** The volumetric soil water layer 1–4, the volumetric soil water layer 1 is 7–10 cm, the volumetric soil water layer 2 is 10–28 cm, the volumetric soil water layer 3 is 28–100 cm, the volumetric soil water layer 4 is 100–289 cm.

Year	Volumetric Soil Water Layer 1/m^3^·m^−3^	Volumetric Soil Water Layer 2/m^3^·m^−3^	Volumetric Soil Water Layer 3/m^3^·m^−3^	Volumetric Soil Water Layer 4/m^3^·m^−3^
1990	0.04	0.12	0.12	0.09
1991	0.04	0.12	0.12	0.09
1992	0.05	0.12	0.12	0.09
1993	0.05	0.13	0.12	0.09
1994	0.06	0.14	0.12	0.09
1995	0.06	0.15	0.13	0.10
1996	0.06	0.16	0.15	0.10
1997	0.05	0.15	0.17	0.10
1998	0.05	0.14	0.16	0.10
1999	0.04	0.14	0.16	0.10
2000	0.04	0.13	0.15	0.10
2001	0.04	0.14	0.15	0.10
2002	0.05	0.14	0.11	0.09
2003	0.05	0.12	0.11	0.09
2004	0.05	0.14	0.11	0.09
2005	0.04	0.12	0.11	0.09
2006	0.04	0.11	0.11	0.09
2007	0.06	0.13	0.11	0.09
2008	0.05	0.15	0.12	0.09
2009	0.05	0.15	0.13	0.09
2010	0.05	0.14	0.13	0.09
2011	0.04	0.12	0.13	0.09
2012	0.04	0.12	0.13	0.09
2013	0.04	0.13	0.13	0.09
2014	0.04	0.12	0.13	0.09
2015	0.06	0.13	0.12	0.09
2016	0.05	0.14	0.12	0.09
2017	0.05	0.05	0.12	0.09
2018	0.05	0.14	0.13	0.09
2019	0.04	0.13	0.13	0.09
2020	0.04	0.13	0.13	0.09
2021	0.04	0.13	0.13	0.09
2022	0.04	0.13	0.12	0.09

## Data Availability

All the data are available from the corresponding author on reasonable request.

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
