# Peer review of "Study of Changes in the Ulan Buh Desert under the Dual Impacts of Desert Farmland Development and Climate Change"

_plants, 2023, doi:10.3390/plants12193510_

Round 1
Reviewer 1 Report
Esteemed Authors,
I extend my gratitude for the privilege of conducting a peer review of your manuscript titled "Study of changes in the Ulan Buh Desert Under the Dual Impacts of Desert Farmland Development and Climate Change." I wish to convey my appreciation for the scientific merit and relevance of your research. However, I must assert that the manuscript necessitates major revisions and refinements to meet the rigorous standards expected in the realm of scholarly publication.
1. In Figure 1, I seek clarification regarding the rationale behind the temporal partitioning denoted by the demarcation point between 2006 and 2007. Additionally, the sudden appearance of the year 2008 as a division point in Section 5 raises questions; is this perhaps a typographical error?
2. It is incumbent upon the authors to elucidate the omission of such partitioning in Figure 3, which appears inconsistent with the approach employed elsewhere in the manuscript.
3. In the interest of enhancing the completeness and utility of your manuscript, I strongly recommend the inclusion of Digital Object Identifiers (DOIs) for all cited references.
4. Section 1 of the manuscript would benefit from a more explicit elucidation of the research's objectives and goals.
5. Furthermore, Section 1 should be augmented to delineate the structural framework of the paper, outlining the content and contributions of each subsequent section.
6. Within the proposed structural framework, Section 2 should encapsulate the comprehensive description of the research methods. I propose that Section 4 be repositioned as Section 2, given its methodological content.
7. In Figure 8, it is advisable to furnish readers with a geographical reference, specifying the research area's location on a map of China or Asia, thereby augmenting the manuscript's accessibility.
8. Subsection 4.2 necessitates substantive revisions. To enhance clarity and transparency, I recommend the construction of a tabular exposition detailing the datasets employed, accompanied by citations to data sources (inclusive of internet links and literature references). Furthermore, elucidate the titles of the datasets utilized within Google Earth Engine (GEE) and specify their spatial resolutions.
9. The utilization of Landsat data requires clarification. Given Landsat's standard provision of two monthly imaging scenes, the paper should elaborate on factors such as cloud cover and the total number of Landsat images incorporated into the analysis.
10. If Google Earth Engine (GEE) was integral to your methodology, I kindly request that you provide an explication of the processing techniques and algorithms employed.
11. The manuscript would benefit from an exhaustive exposition detailing the computational procedures employed in Sections 4.3, 4.4, and 4.5, including references to software tools. Additionally, please elucidate the software packages employed in the creation of the geographical maps.
12. To enhance the manuscript's comprehensibility, I advocate for a thorough review and amendment of notation symbol captions within the mathematical formulas. This rectification will serve to eliminate ambiguity and ensure clarity of interpretation.
13. Section 4.3.4 warrants further elucidation, as it currently lacks clarity regarding the authors' methodology and procedures.
14. A critical aspect of temporal data analysis is the assessment of stationarity or non-stationarity of time series data. I encourage the authors to explicitly address whether such assessments were conducted.
15. In totality, the research methodology requires comprehensive refinement. While the authors' objectives are discernible, the methodologies employed remain inadequately explicated.
16. Figure 2 raises concerns of misleading visual representation. The use of identical red coloration in two separate legends on geographical maps necessitates separation of the figures and the provision of distinct legends for clarity.
17. The rationale for incorporating Table 4 within the "Discussion" section requires explicit justification.
18. To fortify the manuscript's contribution to the field, I propose the inclusion of a quantitative comparative analysis of the data pertaining to the rate of change observed in the Ulan Buh Desert with analogous data from other regions within China, Asia, or globally.
In conclusion, the depth and rigor of your research are unquestionably commendable; however, the manuscript requires substantial revisions to render it suitable for publication in a scientific journal of high repute. I remain available for any further queries or clarification.
Author Response
We thank Editor and anonymous reviewers for their comments. Based on your comment and request, we have made extensive modification on the original manuscript. We have improved the literature review, data analysis and figure quality, with a focus on the analysis and results of NDVI variation and its influencing factors in the Ulan Buh Desert. A document answering every question from the referees is also included, the manuscript has been revised with the assistance of a native English speaker. The manuscript has been significantly improved by addressing the comments. The following are our point-to-point responses to their comments.
I extend my gratitude for the privilege of conducting a peer review of your manuscript titled "Study of changes in the Ulan Buh Desert Under the Dual Impacts of Desert Farmland Development and Climate Change." I wish to convey my appreciation for the scientific merit and relevance of your research. However, I must assert that the manuscript necessitates major revisions and refinements to meet the rigorous standards expected in the realm of scholarly publication.
Response: We would like to express our great appreciation to you for the fact that you think that the results may be of interest to you. Thank you for validating our research. We will take your comments seriously and will make improvements to the manuscript.
- In Figure 1, I seek clarification regarding the rationale behind the temporal partitioning denoted by the demarcation point between 2006 and 2007. Additionally, the sudden appearance of the year 2008 as a division point in Section 5 raises questions; is this perhaps a typographical error?
Response: We have included guidance on the selection of time partitions in section 3.1.1. We sincerely apologize for the previous oversight, and this has now been rectified. For the specific revision, kindly refer to Part 5, Section 2, Line 5. The study period spans from 1990 to 2022, with the year 2007 serving as the midpoint to compare the initial 17 years with the subsequent 16 years.
- It is incumbent upon the authors to elucidate the omission of such partitioning in Figure 3, which appears inconsistent with the approach employed elsewhere in the manuscript.
Response: Implemented, the rationale for not implementing partitioning in Figure 3 has been included in the section following 3.2.1. For more detailed information, please review the annotated section in 3.2.1. Additionally, please refer to 3.2.1.
Figure 4 employs a comprehensive approach to assess NDVI changes in the context of total precipitation and annual mean temperature from 1990 to 2022. This analysis is conducted from both a holistic and a partial standpoint. In general, the NDVI fluctuations observed from 1990 to 2022 exhibit resemblances to the patterns of precipitation changes.
- In the interest of enhancing the completeness and utility of your manuscript, I strongly recommend the inclusion of Digital Object Identifiers (DOIs) for all cited references.
Response: Implemented. We have added DOIs for all cited references.
- Section 1 of the manuscript would benefit from a more explicit elucidation of the research's objectives and goals.
Response: Implemented. We have incorporated the aims and objectives of this investigation. Kindly refer to the opening paragraph:
The primary goal of this study is to offer valuable insights for the future sustainable development of agricultural land, the implementation of measures to mitigate sandstorms, and the enhancement of policies aimed at safeguarding and restoring the natural ecological environment. Furthermore, it aims to bolster warning systems and facilitate the assessment of drought and sandstorm-related risks through the efficient monitoring of precipitation stress and associated vegetation dynamics.
- Furthermore, Section 1 should be augmented to delineate the structural framework of the paper, outlining the content and contributions of each subsequent section.
Response: Implemented. We have integrated and enriched each of the following sections. Please refer to the introductory paragraph:
This study employs Sen-MK trend analysis on NDVI, precipitation, and temperature data to explore temporal trends and their significance from 1990 to 2022. Additionally, it scrutinizes the correlations and significance between NDVI and precipitation as well as temperature. To predict forthcoming trends in vegetation cover, a Hurst analysis is conducted. Lastly, residual analysis is executed to assess the positive and negative influences of human activities on vegetation cover.
- Within the proposed structural framework, Section 2 should encapsulate the comprehensive description of the research methods. I propose that Section 4 be repositioned as Section 2, given its methodological content.
Response: Implemented, section 4 has already been moved to section 2.
- In Figure 8, it is advisable to furnish readers with a geographical reference, specifying the research area's location on a map of China or Asia, thereby augmenting the manuscript's accessibility.
Response: Implemented, we have added the Figure 1.
- Subsection 4.2 necessitates substantive revisions. To enhance clarity and transparency, I recommend the construction of a tabular exposition detailing the datasets employed, accompanied by citations to data sources (inclusive of internet links and literature references). Furthermore, elucidate the titles of the datasets utilized within Google Earth Engine (GEE) and specify their spatial resolutions.
Response: Implemented. We have included Table 1. Data Sources and Resolutions in section 2.2.
- The utilization of Landsat data requires clarification. Given Landsat's standard provision of two monthly imaging scenes, the paper should elaborate on factors such as cloud cover and the total number of Landsat images incorporated into the analysis.
Response: Implemented. We have added a description of the Landsat data in section 2.2:
The NDVI data used in this study were obtained from the cloud computing platform Google Earth Engine (GEE). The data type consists of a long-time series of Landsat 5/7/8 satellite remote sensing data with a spatial resolution of 30m and a temporal resolution of 16 days. Additionally, Landsat remote sensing images were utilized for cloud removal analysis.
- If Google Earth Engine (GEE) was integral to your methodology, I kindly request that you provide an explication of the processing techniques and algorithms employed.
Response: Implemented, we have added an explanation of the processing techniques and algorithms employed in section 2.2:
The primary algorithm utilized involves the use of Google Earth Engine (GEE) to select Landsat 5/7/8 images while effectively masking out cloud cover. Subsequent steps encompass the extraction of the red and near-infrared bands from the selected images, the calculation of NDVI values, image processing, cropping to the specific study area, and the generation of visual representations of the NDVI data. The NDVI data for different years from 1990 to 2022 were generated using the maximum synthesis method on a monthly scale. These data were further processed, including projection transformation, mosaicking, and cropping, to obtain the NDVI dataset for the Ulan Buh Desert and neighboring areas.
- The manuscript would benefit from an exhaustive exposition detailing the computational procedures employed in Sections 4.3, 4.4, and 4.5, including references to software tools. Additionally, please elucidate the software packages employed in the creation of the geographical maps.
Response: Implemented. We have provided a comprehensive explanation of the computational procedures and software packages used in Sections 2.3.1, 2.3.2, 2.3.3, and 2.3.4.
- To enhance the manuscript's comprehensibility, I advocate for a thorough review and amendment of notation symbol captions within the mathematical formulas. This rectification will serve to eliminate ambiguity and ensure clarity of interpretation.
Response: We have conducted a comprehensive review and made amendments to the notation symbol captions within the mathematical formulas.
- Section 4.3.4 warrants further elucidation, as it currently lacks clarity regarding the authors' methodology and procedures.
Response: Implemented. We have enhanced the clarity of the formula explanations and made revisions to the notation symbols within them. Please refer to Section 2.3.4.
In the formulas, NDVI (pre) represents the predicted NDVI value, x1 represents the annual average temperature, x2 represents the annual total precipitation, 'a' represents the constant term, 'b' and 'c' are the regression coefficients, 'σ' denotes the residual value, and NDVI (obs) represents the observed NDVI values.
- A critical aspect of temporal data analysis is the assessment of stationarity or non-stationarity of time series data. I encourage the authors to explicitly address whether such assessments were conducted.
Response: Implemented. We have added an assessment of stationarity. Please refer to the highlighted section in Section 2.3:
All the data underwent stationarity testing using the Augmented Dickey-Fuller (ADF) test within the SPSS software. The results indicate that the data successfully passed the stationarity test. Specifically, after performing first-order differentiation on the NDVI data, its ADF statistic was -8.46366, with a P-value of 0.0078. This P-value is significantly smaller than the 5% significance level, confirming that the sequence is indeed stationary. Regarding the "precipitation" variable, its ADF statistic was -5.994203, with a P-value of 0.0000, indicating stationarity as the P-value is very close to zero and significantly less than the 5% significance level. For the "temperature" variable, its ADF statistic was -3.218903, with a P-value of 0.0280. Although the P-value is not extremely small, it is still less than the 5% significance level, suggesting that the sequence is stationary. In conclusion, based on these results, it can be firmly asserted that the sequence exhibits strong stability after first-order differencing, demonstrating a first-order integer stationarity.
- In totality, the research methodology requires comprehensive refinement. While the authors' objectives are discernible, the methodologies employed remain inadequately explicated.
Response: Implemented. We have provided detailed explanations for each method and introduced the application of the article's structural approach in the first paragraph of Section 2.
- Figure 2 raises concerns of misleading visual representation. The use of identical red coloration in two separate legends on geographical maps necessitates separation of the figures and the provision of distinct legends for clarity.
Response: Implemented, we have changed the same color in the legend.
- The rationale for incorporating Table 4 within the "Discussion" section requires explicit justification.
Response: Implemented. We have provided a comprehensive explanation of Table 5 in the preceding paragraph. This table presents soil moisture data at varying depths spanning from 1990 to 2022. The dataset facilitates an in-depth examination of the long-term fluctuations in soil moisture, particularly in light of the impacts stemming from agricultural irrigation practices and alterations in vegetation cover.
- To fortify the manuscript's contribution to the field, I propose the inclusion of a quantitative comparative analysis of the data pertaining to the rate of change observed in the Ulan Buh Desert with analogous data from other regions within China, Asia, or globally.
Response: Implemented. We have incorporated a comparative analysis between the Ulan Buh Desert and the Mongolian Plateau. Please refer to the final portion of the second paragraph in Section 4:
We have compared the rate of NDVI increase between the Ulan Buh Desert and the Mongolian Plateau. The Mongolian Plateau witnessed substantial growth from 2010 to 2019, exhibiting a regional increase of 9.08% in comparison to the period from 2000 to 2009. Conversely, in the Ulan Buh Desert, a significant increase of 7.30% occurred between 2007 and 2022 when compared to the period from 1990 to 2006. Overall, the growth rate of the Ulan Buh Desert and the plateau is relatively gradual. The disparity between the NDVI trend of the Ulan Buh Desert and the predicted H value of the Mongolian Plateau is only 0.0119, suggesting a consistent future trend. The research area exhibits correlations with both precipitation and temperature, while the Mongolian Plateau is only correlated with precipitation.
In conclusion, the depth and rigor of your research are unquestionably commendable; however, the manuscript requires substantial revisions to render it suitable for publication in a scientific journal of high repute. I remain available for any further queries or clarification.
Response: We thank anonymous reviewer for recognition of my research content and depth, as well as for your valuable suggestions. I will carefully revise the manuscript for publication in scientific journals. Thank you for your assistance.

Reviewer 2 Report
I have carefully read the manuscript titled “Study of Changes in the Ulan Buh Desert Under the Dual Im-pacts of Desert Farmland Development and Climate Change”. NDVI is an effective approach to addressing the impact of climate change. The authors did a solid literature review and indicated their work's goals and applications carefully. This should have been the first step and included in its local application. This ms represents a comprehensive survey of data, and analysis and has effectively addressed important gaps in our current state of knowledge. That being said, I have some concerns that need to be raised, the presentations of figures/ figure legends and discussions need more details and connections.
In particular, most of the figure legends needed to be more concise. The authors have had to focus on certain aspects of the analysis goal and several climate factors they are discussing, and they omit a range of indications and processes (such as Figure2 (d)) that are critical in the current discussion and debates. They need to lay out the limitations of their study or include the prospects more precisely in the discussion and conclusion paragraphs. On the other hand, in the precipitation, it not convincing that the factor remains relatively stable (see Figure4 (c)). I recommend incorporating and revising these into the ms before the Journal accepts it.
I hope these comments are useful.
I have carefully read the manuscript titled “Study of Changes in the Ulan Buh Desert Under the Dual Im-pacts of Desert Farmland Development and Climate Change”. NDVI is an effective approach to addressing the impact of climate change. The authors did a solid literature review and indicated their work's goals and applications carefully. This should have been the first step and included in its local application. This ms represents a comprehensive survey of data, and analysis and has effectively addressed important gaps in our current state of knowledge. That being said, I have some concerns that need to be raised, the presentations of figures/ figure legends and discussions need more details and connections.
In particular, most of the figure legends needed to be more concise. The authors have had to focus on certain aspects of the analysis goal and several climate factors they are discussing, and they omit a range of indications and processes (such as Figure2 (d)) that are critical in the current discussion and debates. They need to lay out the limitations of their study or include the prospects more precisely in the discussion and conclusion paragraphs. On the other hand, in the precipitation, it not convincing that the factor remains relatively stable (see Figure4 (c)). I recommend incorporating and revising these into the ms before the Journal accepts it.
I hope these comments are useful.
Author Response
We thank Editor and an anonymous reviewer for their comments. Based on your comment and request, we have made extensive modification on the original manuscript. We have improved the literature review, data analysis and figure quality, with a focus on the analysis and results of NDVI variation and its influencing factors in the Ulan Buh Desert. A document answering every question from the referees is also included, the manuscript has been revised with the assistance of a native English speaker. The manuscript has been significantly improved by addressing the comments. The following are our point-to-point responses to their comments.
I have carefully read the manuscript titled “Study of Changes in the Ulan Buh Desert Under the Dual Impacts of Desert Farmland Development and Climate Change”. NDVI is an effective approach to addressing the impact of climate change. The authors did a solid literature review and indicated their work's goals and applications carefully. This should have been the first step and included in its local application. This ms represents a comprehensive survey of data, and analysis and has effectively addressed important gaps in our current state of knowledge. That being said, I have some concerns that need to be raised, the presentations of figures/ figure legends and discussions need more details and connections.
Response: Thank you for your recognition of the manuscript. We appreciate your acknowledgement of the data. We carefully improve the figures, tables, and legends to ensure a more rigorous and meticulous manuscript. We have revised the legends for Figure 6 and Figure 7 to enhance clarity. In Figure 6, we have consolidated the legends to streamline the presentation in Section 3.2.2. In Figure 7, we have eliminated the legend 'Study Area' to create a more concise and straightforward legend in Section 3.3.
In particular, most of the figure legends needed to be more concise. The authors have had to focus on certain aspects of the analysis goal and several climate factors they are discussing, and they omit a range of indications and processes (such as Figure2 (d)) that are critical in the current discussion and debates. They need to lay out the limitations of their study or include the prospects more precisely in the discussion and conclusion paragraphs. On the other hand, in the precipitation, it not convincing that the factor remains relatively stable (see Figure4 (c)). I recommend incorporating and revising these into the ms before the Journal accepts it.
Response: Implemented. We made changes in these four areas:
- Figure Legends and Figure 4 (d) Analysis: Initially, we improved the legends of some figures, including Figure 6 and Figure 7. Additionally, we've incorporated a detailed analysis concerning Figure 4 (d). For an in-depth explanation, please consult the second paragraph of Section 3.1.2.
- Stationarity of Variables: Furthermore, we assessed the stationarity of annual precipitation, average annual temperature, and NDVI through Augmented Dickey-Fuller (ADF) tests. These variables successfully met the criteria for stationarity. Refer to the marked section at the conclusion of Section 2.3.1 for specific details.
- Discussion on Limitations and Future Research: We've included a discussion within the conclusion section to outline the study's limitations and present prospects for future research. Explore the concluding part of Section 4 for comprehensive insights.
- Consideration of Additional Climate Factors: Notably, this study has primarily focused on two climate factors: annual mean temperature and annual total precipitation, while evaluating their impact on vegetation NDVI. Future research endeavors should encompass a more extensive array of meteorological factors. Moreover, it's imperative to integrate land-use categories and scrutinize NDVI alterations under various land cover scenarios. Subsequent studies should also factor in local policies, harmoniously amalgamate theory and practice, and offer an in-depth exploration of the reaction of vegetation to climate and human activities. The correlation linking NDVI, climate variables, and vegetation is inherently dynamic and may manifest variations across distinct temporal scales. To achieve a comprehensive understanding, careful consideration should be extended to seasonal or interannual fluctuations. In forthcoming investigations, the application of machine learning algorithms and modeling techniques can serve as invaluable tools for uncovering concealed patterns and projecting future relationships between NDVI, climate, and vegetation. These methodologies have the potential to facilitate informed decision-making within domains like land management, resource allocation, and ecosystem conservation.
I hope these comments are useful.
Response: I think your suggestion has been very helpful to me. Thank you for your suggestion, which has improved the professionalism of the manuscript.

Round 2
Reviewer 1 Report
I am grateful to the authors for clear and detailed answers. However, I still have some questions, the answers to which will improve the quality of the article.
1. On the map of countries in Figure 1 (upper left corner) specify the coordinate grid.
2. The research methodology has become a little clearer, but not completely. As a reader, I should be able to fully reproduce your research for the application of the methodology in any other region of the world. I can't do that at the moment. I am not satisfied with the answers to questions 9, 11. Have you used this collection - "LANDSAT/LC08/C 02/T2_TO"? Or another one? Or maybe other data? How you handled them.
3. Specify the lengths (number of processed points) the time series you are processing for all variables.
Author Response
We thank Editor and an anonymous reviewer for their comments. Based on your comment and request, we have made fine-tuned modification on the second revised manuscript. A document answering every question from the referees is also included, the manuscript has been revised with the assistance of a native English speaker. The manuscript has been significantly improved by addressing the comments. The following are our point-to-point responses to their comments.
I am grateful to the authors for clear and detailed answers. However, I still have some questions, the answers to which will improve the quality of the article.
Response: Thank you for your feedback. I will continue to make improvements based on your valuable suggestions, as they are important for improving the quality of the article. I appreciate your comments.
- On the map of countries in Figure 1 (upper left corner) specify the coordinate grid.
Response: Implemented. We have added the specify the coordinate grid of Figure 1 (upper left corner).
- The research methodology has become a little clearer, but not completely. As a reader, I should be able to fully reproduce your research for the application of the methodology in any other region of the world. I can't do that at the moment. I am not satisfied with the answers to questions 9, 11. Have you used this collection - "LANDSAT/LC08/C 02/T2_TO"? Or another one? Or maybe other data? How you handled them.
Response: Implemented. As the author, we will strive to make my research methodology clear and straightforward. We used three datasets, namely LANDSAT/LT05/C02/T1_L2, LANDSAT/LE07/C02/T1_L2, and LANDSAT/LC08/C02/T1_L2, which are included in the revised section 2.2:
We performed cloud removal processing on the Google Earth Engine (GEE) platform. We applied scaling coefficients to the appropriate bands, replaced them with the scaled bands, and applied a cloud mask. We selected the desired time frame for the data and synthesized the NDVI values using the mean. We have also added a detailed description of the processing data. Please refer to the section 2.3.1:
The NDVI, precipitation, and temperature data for the periods 1990-2006 and 2007-2022 are separately input into the MATLAB Starter Application as raster datasets. The code is then run to compute the corresponding trends and perform the Mann-Kendall test for each individual year. In the ArcGIS 10.8 software, masked extraction is performed using the results obtained. The trend maps are generated directly from the output of the code, while the significance results are categorized based on the trend maps. finally, the figures are generated.
Please refer to the Section 2.3.2:
In the RStudio software, you can import the excel-formatted data for NDVI, precipitation, and temperature for the periods 1990-2006 and 2007-2022. Prior to that, you need to download the necessary packages. Using the correlation formula and specific code, a correlation scatterplot can be generated to visualize the relationships.
Please refer to the Section 2.3.3:
To perform the Hurst trend prediction, the MATLAB Starter Application requires the loading of raster data for the years 1990-2022. The appropriate code is in-putted to predict future trends. These predicted trend images for the NDVI are then integrated, and classification and masked extraction are performed using ArcGIS 10.8 software.
Please refer to the Section 2.3.4:
For the residual analysis, the MATLAB Starter Application is used to load NDVI raster images corresponding to the periods 1990-2006 and 2007-2022. The code is executed to compute the predicted NDVI values, residuals, one-dimensional linear regression trends, and the impacts of climate change and human activities for each year. Finally, the processed raster images are subjected to mask extraction and classification using ArcGIS 10.8.
- Specify the lengths (number of processed points) the time series you are processing for all variables.
Response: Implemented. Please refer to the sentences and table 1 of Section 2.2.
The time series for NDVI, precipitation, and temperature span from 1990 to 2022, covering 32 years of continuous data. The DEM data represent only one year.
|
Data Type |
Dataset |
Spatial Resolution |
Length of time |
|
NDVI (1990-2011) |
LANDSAT/LT05/C02/T1_L2 |
30m |
32 |
|
NDVI (2012-2013) |
LANDSAT/LE07/C02/T1_L2 |
||
|
NDVI (2014-2022) |
LANDSAT/LC08/C02/T1_L2 |
||
|
Precipitation |
http://data.cma.cn/ |
1km |
|
|
Temperature |
http://data.cma.cn/ |
1km |
|
|
DEM |
NASA/NASADEM_HGT/001 |
30m |
1 |

Round 3
Reviewer 1 Report
Esteemed authors,
I have received responses to my inquiries, and I find them satisfactory in addressing all but one particular question. To my understanding, the calculation of the Hurst index was based on a dataset spanning 32 years. However, the methodology used for calculating the Hurst index remains unspecified. It is crucial to elucidate whether the index was computed based on 32-yearly averaged values or derived from the entire set of NDVI values without annual averaging. Clarification on this matter is imperative for a comprehensive understanding.
The time series is insufficiently long, consisting of only 32 yearly data points, making it challenging to conduct a trustworthy scaling analysis and accurately interpret the estimated Hurst exponents. A lower limit for the size of the sample is usually N=2^8=256 (e.g., Crevecoeur et al, 2010) while the convergence of the estimation to the real value is obtained asymptotically with increasing N.
Crevecoeur F, Bollens B, Detrembleur C, Lejeune TM. Towards a "gold-standard" approach to address the presence of long-range auto-correlation in physiological time series. J Neurosci Methods. 2010 Sep 30;192(1):163-72. doi: 10.1016/j.jneumeth.2010.07.017.
Author Response
We thank Editor and a reviewer for their comments. Based on your comment and request, we have made some modification on the manuscript. We have improved the method of Hurst index and revised the order of the references and the overall. The manuscript has been significantly improved by addressing the comments. The following are our point-to-point responses to their comments.
I have received responses to my inquiries, and I find them satisfactory in addressing all but one particular question. To my understanding, the calculation of the Hurst index was based on a dataset spanning 32 years. However, the methodology used for calculating the Hurst index remains unspecified. It is crucial to elucidate whether the index was computed based on 32-yearly averaged values or derived from the entire set of NDVI values without annual averaging. Clarification on this matter is imperative for a comprehensive understanding.
Response: Thank you very much for your valuable suggestions and insights. We will now provide a detailed explanation of the specific methodology used for calculating the Hurst index. It's important to note that the Hurst index was calculated directly from the entire set of NDVI values without annual averaging.
The time series is insufficiently long, consisting of only 32 yearly data points, making it challenging to conduct a trustworthy scaling analysis and accurately interpret the estimated Hurst exponents. A lower limit for the size of the sample is usually N=2^8=256 (e.g., Crevecoeur et al, 2010) while the convergence of the estimation to the real value is obtained asymptotically with increasing N.
Response: Implemented. We added the We have added a portion of the supplement in section 2.3.3.
For the analysis of the Hurst index, this study analysed 32 years of NDVI data, with 13 data points per year, for a total of 416 data points. In total, more than 256 data points were analysed. The results are highly credible as the data points meet the quantitative requirements of the Crevecoeur et al. study.

Round 4
Reviewer 1 Report
Accept in present form